# Near-Infrared Photothermally Enhanced Photo-Oxygenation for Inhibition of Amyloid-β Aggregation Based on RVG-Conjugated Porphyrinic Metal–Organic Framework and Indocyanine Green Nanoplatform

**DOI:** 10.3390/ijms231810885

**Published:** 2022-09-17

**Authors:** Jiuhai Wang, Yutian Gu, Xu Liu, Yadi Fan, Yu Zhang, Changqing Yi, Changming Cheng, Mo Yang

**Affiliations:** 1Department of Biomedical Engineering, The Hong Kong Polytechnic University, Hong Kong 999077, China; 2Department of Mechanical and Automotive Engineering, Royal Melbourne Institute of Technology University, Melbourne, VIC 3000, Australia; 3Key Laboratory of Sensing Technology and Biomedical Instruments Guangdong, School of Biomedical Engineering, Sun Yat-Sen University, Guangzhou 510006, China; 4Institute of Nuclear Physics and Chemistry, China Academy of Engineering Physics (CAEP), Mianyang 621900, China

**Keywords:** amyloid-β, neurodegenerative diseases, central nervous system (CNS), Alzheimer’s disease, metal–organic framework, near-infrared phototherapy, brain-targeting, photo-oxygenation, porphyrinic nanoparticles

## Abstract

Amyloid aggregation is associated with many neurodegenerative diseases such as Alzheimer’s disease (AD). The current technologies using phototherapy for amyloid inhibition are usually photodynamic approaches based on evidence that reactive oxygen species can inhibit Aβ aggregation. Herein, we report a novel combinational photothermally assisted photo-oxygenation treatment based on a nano-platform of the brain-targeting peptide RVG conjugated with the 2D porphyrinic PCN−222 metal–organic framework and indocyanine green (PCN−222@ICG@RVG) with enhanced photo-inhibition in Alzheimer’s Aβ aggregation. A photothermally assisted photo-oxygenation treatment based on PCN@ICG could largely enhance the photo-inhibition effect on Aβ_42_ aggregation and lead to much lower neurotoxicity upon near-infrared (NIR) irradiation at 808 nm compared with a single modality of photo-treatment in both cell-free and in vitro experiments. Generally, local photothermal heat increases the instability of Aβ aggregates and keeps Aβ in the status of monomers, which facilitates the photo-oxygenation process of generating oxidized Aβ monomers with low aggregation capability. In addition, combined with the brain-targeting peptide RVG, the PCN−222@ICG@RVG nanoprobe shows high permeability of the human blood–brain barrier (BBB) on a human brain-on-a-chip platform. The ex vivo study also demonstrates that NIR-activated PCN−222@ICG@RVG could efficiently dissemble Aβ plaques. Our work suggests that the combination of photothermal treatment with photo-oxygenation can synergistically enhance the inhibition of Aβ aggregation, which may boost NIR-based combinational phototherapy of AD in the future.

## 1. Introduction

Alzheimer’s disease (AD) is a severe neurodegenerative disorder with over 47 million cases in the world [1,2]. AD is a major cause of disability and dependency among the elderly. Neurodegeneration is caused by a structural and/or functional loss of neurons in the central nervous system (CNS), leading to the impairment and dysfunction of the corresponding motor, autonomic, and cognitive nervous systems in the brain [3,4,5]. Studies have found that several neuroactive compounds and their signaling pathways, through various types of receptors such as glutamate and γ-aminobutyric acid (GABA), are important in brain homeostasis [6,7]. Additional studies have also revealed that the disconnection between microglial and other brain cells, including neurons, may be involved in the pathomechanism of AD [8,9,10]. Neuroinflammation can damage neurons and result in Aβ aggregation in the brain, so this anti-inflammatory has become a promising approach for AD treatment [11,12,13,14,15,16]. Increasing evidence suggests that one of the causes of this neurodegeneration is the overproduction and misfolding of the extracellular amyloid-beta (Aβ) peptide to neurotoxic Aβ aggregates such as oligomers and fibrils with β-sheet-rich structures [17,18,19]. One potential treatment option is to suppress Aβ aggregation so as to decrease synaptic neurotoxicity [20]. Many efforts have been made in the last decade to develop various anti-Aβ-targeting molecules capable of suppressing Aβ aggregation and neurotoxicity, which opens a new era for the preventative treatment of AD at the early pre-dementia phase [21]. However, the majority of the above inhibitors show low inhibition capability [22]. Moreover, these anti-Aβ-targeting molecules suffer from rapid degradation in plasma and poor blood–brain barrier (BBB) permeability [23].

Near-infrared (NIR)-based phototherapy is an attractive option for curing local diseases owing to its temporal and spatial controllability and reduced side-effects [24,25]. Recently, it has been found that the chemical transformation of native Aβ to oxygenated forms via photo-oxygenation could successfully suppress Aβ aggregation and neurotoxicity [26,27,28]. Many materials have been reported to be photoresponsive, and can transfer photo-energy to other forms of energy such as electrical energy and chemical energy [29,30,31]. Many kinds of nanomaterial-based photosensitizers have been used for the inhibition of Aβ aggregation in the visible-light window (including carbon nanodots [32] and graphitic carbon nitride nanosheets (g−C_3_N_4_) [33]), and in the near-infrared light (NIR) window (including nanocomposites of Yb/Er-co-doped NaYF4 upconversion nanoparticles (UCNP) and RB [34], and black phosphorus (BP)) [35]. Although the inhibition of Aβ aggregation via localized photo-oxygenation has been well studied, the relatively low photo-inhibition efficiency limits its application.

In addition to photo-oxygenation, the photothermal inhibition of amyloid aggregation has recently become another promising approach to AD therapy [36]. Aβ aggregate formation is highly dependent on temperature, so local hyperthermia could suppress the formation of misfolded Aβ aggregates [37]. Studies have reported that photothermal treatment could locally and remotely heat and inhibit Aβ aggregation based on graphene oxide (GO) and tungsten disulfide (WS_2_) nanosheets via near-infrared laser (NIR) irradiation [36,37]. However, photo-oxygenation probes suffer from moderate efficiency of the inhibition of Aβ aggregation. In addition, the photothermal inhibition effect is temporary. Last but not least, the current nanoprobes also lack human blood–brain barrier (BBB) permeability, which limits their future applications. Therefore, the combination of photothermal heating and photo-oxygenation treatment in one single nano-system with good BBB permeability, under moderate NIR light irradiation, may be a good synthetic strategy to suppress Aβ aggregation and neurotoxicity with enhanced therapeutic efficacy. Near-infrared light in the range of 600~1000 nm has been used to penetrate the skull and reach the shallower brain regions (such as the cerebral cortex) for light stimulation of the brain [38,39], where AD amyloid accumulation usually occurs [40]. An in vivo situation would be more complicated compared to an ex vivo experiment due to the decay of NIR intensity by the human skull and the biological tissues surrounding it. Therefore, the largely enhanced Aβ aggregation inhibition efficacy under moderate NIR laser intensity via a combinational phototherapy approach is important for future in vivo phototherapy of AD.

The current development of therapeutic nanoprobes for AD is limited by a lack of predictive models to reliably assess the BBB permeability of therapeutic drugs [41,42]. In vivo animal models of AD suffer from high cost and labor intensity, and ethical concerns [43]. Moreover, the transport functions of the BBB in animal models may be significantly different from those of human patients. Organ-on-a-chip technology replicates organ-level functions and the cellular microenvironment, and can be good models to study the physiological responses of therapeutic probes at a low cost [44,45,46,47,48,49,50,51]. It is of high importance to develop a microfluidic organ-on-a-chip with the human blood–brain barrier which can mimic the human brain microenvironment of Alzheimer’s disease, with neurotoxicity induced by Aβ, as pre-clinical tools to study brain-targeting therapeutic probes.

Here, we reported near-infrared (NIR) light-induced combinational photo-inhibition of Aβ aggregation via the hybrid brain-targeting peptide rabies virus glycoprotein (RVG) conjugated with the porphyrinic metal–organic framework and indocyanine green nanoprobe (PCN−222@ICG@RVG). The RVG29 peptide is a protein that specifically binds the nicotinic acetylcholine receptor (nAchR), which is a receptor widely expressed on the surface of brain endothelial cells [52,53,54,55,56]. The hybrid PCN−222@ICG@RVG nanoprobe processes enhanced the inhibition efficiency of Aβ aggregation and improves neurotoxicity attenuation under NIR light exposure compared with a single modality of photo-treatment; this is attributed to the combined effect of the thermodynamic instability of Aβ aggregates caused by local photothermal heating, and the inhibition of Aβ monomer aggregation induced by photo-oxygenation (Figure 1). The hybrid nanoprobe has proven its good human BBB permeability in a microfluidic brain-on-a-chip model under a continuous flow that mimics the in vivo brain microenvironment of Alzheimer’s disease. The enhanced dissociation of Aβ plaques via NIR-stimulated hybrid nanoprobes is also observed in ex vivo studies. This research highlights the promising approach of combinational NIR-based phototherapy in halting the progression of Alzheimer’s disease in the future.

## 2. Results

### 2.1. Preparation and Identification of PCN-222@ICG

Metal–organic frameworks (MOFs) are emerging nanomaterials with unique mechanical, electrical, and chemical properties that have been used in various biomedical applications [57,58,59]. In this study, PCN−222@ICG nanoprobes were prepared using a two-step method including the solvothermal synthesis of a 2D porphyrinic MOF PCN-222 nanosheet and the immobilization of ICG onto MOF nanosheets via electrostatic adsorption. Compared to 3D bulk MOF crystals, 2D MOF nanosheets have a higher percentage of exposed catalytically active sites and a larger surface area, which facilitates the loading of ICG agents and the diffusion of molecular oxygen. The as-synthesized PCN−222 nanosheet exhibited an ultrathin 2D crystal structure (Appendix A). The PCN−222 nanosheet did not change its 2D morphology after the electrostatic adsorption of ICG in the TEM images (Figure 1a). Dynamic light scattering (DLS) measurement indicated that the DLS size of PCN−222@ICG was around 120 nm (Figure 1b). The UV-vis absorption spectra showed a significant increase in wavelength range between 700 and 900 nm after ICG adsorption, demonstrating the successful loading of ICG on the PCN−222 nanosheet (Figure 1c). The powder X-ray diffraction (PXRD) pattern of both PCN and PCN@ICG displayed three obvious peaks assigned to (0, 0, 1), (2, 0, 1), and (3, 1, 1), which were consistent with the standard XRD spectrum of PCN−222 (Figure 1d). The N_2_ absorption/desorption spectrum demonstrated the good porous structure of PCN−222@ICG, which was beneficial for molecular oxygen diffusion (Appendix A). The porosity of PCN−222@ICG is 198.74 m^2^ g^−1^ and the average pore size is 38.8321 Å. Zeta potential measurement showed that the PCN nanosheet was positively charged and ICG was negatively charged. The loading of ICG on the PCN−222 nanosheet led to the negative surface charge of PCN−222@ICG (Figure 1e). The color of the nanoprobes dispersed in water changed from red to green, which also proved the successful loading of ICG (Figure 1e inset).

### 2.2. Characterization of Photothermal Effect of PCN@ICG

To examine the photothermal effect of PCN−222@ICG, an 808 nm laser was employed to irradiate the nanoprobes and the photothermal effect was measured using a thermal camera (Figure 2). PCN−222@ICG with various concentrations (0.025 mg mL^−1^, 0.05 mg mL^−1^, 0.1 mg mL^−1^, and 0.2 mg mL^−1^), and PCN-222 (0.2 mg mL^−1^) and DI water as control groups were irradiated under 808 nm NIR laser excitation with the same power density of 0.6 W cm^−2^ (Figure 2a). Under continuous laser irradiation of 5 min, PCN−222@ICG groups with various concentrations showed a quick temperature increase with an obvious concentration dependence (Figure 2a,c). For example, the temperature of PCN−222@ICG with a low concentration of 0.025 mg mL^−1^ rapidly increased from 25 °C to 45 °C after 5 min of laser irradiation. In contrast, the PCN−222-alone and DI water groups did not show obvious temperature changes after 5 min of laser irradiation. Moreover, PCN−222@ICG also showed a power-density dependence within 5 min of laser irradiation (Figure 2b,c). The photothermal conversion efficiency of PCN−222@ICG (0.025 mg mL^−1^) was calculated to be 71.2% according to the photothermal heating/dissipation cycle (Figure 2d). It was observed that the temperature of the nanoprobe solution quickly dropped to room temperature after the laser source was turned off, suggesting excellent thermal conductivity of the hybrid nanoprobe. The photothermal stability was further evaluated by reversibly heating and cooling the PCN−222@ICG solution (0.025 mg mL^−1^) at a power density of 0.6 W cm^−2^ for five cycles (Figure 2e). The maximum temperatures of the nanoprobes remained nearly the same for five cycles, which demonstrated the good photothermal stability of PCN−222@ICG. The above results evidenced that PCN−222@ICG had good photostability under NIR laser irradiation. To further investigate the photostability of the nanoprobes, PCN−222@ICG was irradiated under NIR laser irradiation (0.6 W cm^−2^) for 30 min and its UV-vis absorbance and zeta potential were recorded. As shown in Appendix A, there was no significant change in the absorption peaks or zeta potential values, which evidenced the photostability of the hybrid nanoprobe. The above results demonstrate that the hybrid PCN−222@ICG nanoprobe could produce remote and localized heat in an efficient manner under NIR laser irradiation.

### 2.3. Characterization of ^1^O_2_ Generation of PCN-222@ICG

Zirconium-porphyrinic MOFs have strong photon absorption and are able to transform ^3^O_2_ to ^1^O_2_ upon NIR light irradiation. To verify the generation of ^1^O_2_ species, a 1,3-diphenyl-isobenzofuran (DPBF) assay was used to measure the singlet oxygen production of the synthesized PCN−222@ICG nanoprobe. The absorption of DBPF at the characteristic 414 nm wavelength decreased to one fifth in the presence of the PCN−222@ICG nanoprobe under moderate NIR light irradiation (808 nm, 0.6 W cm^−2^) within ten minutes, revealing the strong generation of ^1^O_2_ species (Figure 3a). A DPBF assay was then used to compare the singlet oxygen generation capability between PCN−222@ICG and PCN−222 under moderate NIR light irradiation (808 nm, 0.6 W cm^−2^). Obviously, PCN−222@ICG showed an enhanced capability of single oxygen generation compared with PCN−222 only (Figure 3b). Electron spin resonance (ESR) was then used to compare the singlet oxygen generation capability between PCN−222@ICG and PCN−222. Here, 4-oxo-2,2,6,6-tetramethylpiperidine (4-oxo-TEMP) was used to react with ^1^O_2_ to yield the stable nitroxide radical 4-oxo-TEMPO, which could be measured through ESR. As shown in Figure 3c, the characteristic ^1^O_2_-induced TEMPO signal was observed in the ESR spectra under NIR light irradiation and its intensity increased with the increase in irradiation time. The ESR signal intensity of PCN−222@ICG was obviously higher than that of PCN−222 under NIR irradiation at various time points, which further demonstrated the enhanced singlet oxygen generation capability.

### 2.4. Enhanced Photo-Induced Inhibition of Aβ_42_ Aggregation based on PCN-222@ICG

Dynamic Light Scattering (DLS) and Transmission Electronic Microscopy (TEM) were used to evaluate the Aβ aggregation dynamic process in a 24 h time-dependent inhibition study. DLS measurement showed that untreated Aβ_42_ monomers quickly increased in size from ~10 nm to ~800 nm within 6 h, and then, further increased to ~1000 nm at 24 h (Figure 4a). Photo-activated PCN−222 with NIR irradiation (0.025 mg mL^−1^, 0.6 W cm^−2^, 30 min) showed a certain inhibition effect on Aβ_42_ aggregation with a size increase to ~200 nm at 24 h. In contrast, photo-activated PCN−222@ICG with NIR irradiation (0.025 mg mL^−1^, 0.6 W cm^−2^, 30 min) showed a strong inhibition effect on Aβ_42_ aggregation with a size increase to only ~90 nm at 24 h. TEM images also clearly showed distinct differences in morphology among Aβ_42_ monomer samples incubated in the absence of nanoprobes, and in the presence of NIR-activated PCN−222 and photo-activated PCN−222@ICG nanoprobes. The Aβ_42_ monomer group in the absence of nanoprobes at 24 h clearly showed morphology of the amyloid fibril mesh network (Figure 4b), and Aβ_42_ with photo-activated PCN−222 showed a low degree of formation of amyloid fibrils (Figure 4c). In contrast, the Aβ_42_ monomer group incubated with photo-activated PCN−222@ICG for 24 h could more efficiently prevent Aβ_42_ from aggregating to a larger size without forming amyloid fibrils (Figure 4d). In contrast, there was no obvious aggregation of the Aβ_42_ monomer at 0h (Appendix A). Our studies demonstrated that NIR-activated PCN−222@ICG was able to more noticeably inhibit Aβ_42_ aggregation than PCN−222 alone by controlling the size of Aβ structures to sub-100 nm. To further analyze the enhanced effect of PCN−222@ICG on inhibiting the aggregation of Aβ_42_ monomers to high-degree structures, circular dichroism (CD) was used to evaluate the secondary structure of the Aβ_42_ aggregate. The untreated Aβ_42_ sample after 24 h of incubation at 37 °C showed a characteristic peak at 216 nm in the CD spectrum, which was a typical profile of the β-sheet-rich secondary structure of amyloids (Figure 4e). In contrast, the Aβ_42_ sample treated with photo-activated PCN−222 and PCN−222@ICG with NIR irradiation (0.025 mg mL^−1^, 0.6 W cm^−2^, 30 min), followed by 24 h of incubation at 37 °C, showed a low peak amplitude at 216 nm, indicating an inhibition effect on the formation of the secondary structure of amyloids. Native polyacrylamide gel electrophoresis (PAGE) analysis was also conducted to compare the inhibition effect of PCN−222 and PCN−222@ICG on the formation of high-degree structures of amyloids after incubation for 24 h. In Figure 4f, lane 1 represents untreated Aβ_42_, lane 2 represents PCN−222 with NIR laser irradiation, lane 3 represents PCN−222@ICG without NIR laser irradiation, and lane 4 represents PCN−222@ICG with NIR laser irradiation. Compared with the other groups from lane 1 to lane 3, which mostly showed aggregation of high-molecular-weight Aβ_42_ oligomers and fibrils, lane 4 of PCN−222@ICG with NIR laser irradiation showed multiple bands of low-order structures, including monomers and oligomers (Figure 4f). This indicated that photo-excited PCN−222@ICG could efficiently inhibit the conversion of Aβ_42_ monomers to high-order structures. The FT-IR spectrum of Aβ_42_ incubated for 24 h in the absence of nanoprobes revealed a characteristic peak at 1636 cm^−1^, which clearly indicated the formation β-sheet structure (Appendix A). In contrast, the Aβ_42_ monomer group in the presence of PCN−222@ICG under NIR irradiation did not show a significant peak at 1636 cm^−1^, indicating an efficient inhibition effect on β-sheet structure formation.

### 2.5. Photothermal Enhancement Effect on Photo-Oxygenation-Based Inhibition of Aβ_42_ Aggregation

To investigate the photothermal enhancement effect of PCN−222@ICG nanoprobes on the inhibition of Aβ_42_ aggregation, PCN−222 (0.025 mg mL^−1^) and PCN−222@ICG (0.025 mg mL^−1^) in a constant-temperature water bath at 25 °C were irradiated with NIR light (808 nm, 0.6 W cm^−2^, 10 min) in the presence of Aβ_42_ monomers (25 µM) (Figure 5). Here, the PCN−222@ICG group in a constant-temperature water bath at 25 °C was used to exclude the photothermal effect. The group with Aβ_42_ monomers only (25 µM), without nanoprobes under NIR irradiation, was designated as “Native” for the control group. After NIR irradiation, various groups were further incubated at 37 °C for 48 h to monitor the aggregation degree of the Aβ_42_ monomers. The Aβ_42_ fibrillation kinetics were monitored using a commonly used Thioflavin T (ThT) assay. ThT is an extrinsic fluorescent dye that specifically binds to β-sheet-rich amyloid structures with a significant fluorescence increase at a wavelength of 485 nm [60]. The results show that the ThT fluorescence signal of native Aβ_42_ without treatment with NIR-activated nanoprobes drastically increased to a plateau after the first 24 h of incubation, indicating the formation of an Aβ_42_ aggregate (Figure 5a). In contrast, the groups of p-MOF and photo-excited PCN−222@ICG excluding the photothermal effect (no PT) and photo-excited PCN−222@ICG all showed inhibition of Aβ_42_ aggregation. Obviously, the PCN−222@ICG group under NIR irradiation showed the lowest ThT fluorescence signal. Figure 5b compares the relative ThT fluorescence signal change of the above groups after 48 h of incubation. Here, the relative ThT fluorescence signal change of native Aβ_42_ without nanoprobes under NIR irradiation is defined as 100%. It was observed that NIR-activated PCN−222@ICG excluding the photothermal effect (no PT) showed a relative fluorescence signal of 26.9%. In contrast, photo-excited PCN−222@ICG showed a much lower relative fluorescence signal of 9.6%, which demonstrated that the photothermal effect played an important role during the photo-oxygenation-based inhibition of Aβ_42_ aggregation. Compared with the PCN−222@ICG group excluding the photothermal effect (no PT), this large decrease in the relative fluorescence signal of the PCN−222@ICG group demonstrated further enhanced inhibition of Aβ_42_ aggregation due to the photothermal effect.

The photothermal effect on the enhanced inhibition of Aβ_42_ aggregation can be explained as follows. Under NIR irradiation, the local photothermal heat generated by PCN−222@ICG will cause instability of the Aβ aggregate and keep most of the Aβ in the status of monomers. At the same time, Aβ_42_ monomers are easily photo-oxygenated by local singlet oxygen generated from PCN−222@ICG, and photo-oxygenated Aβ monomers have much lower aggregation capability compared with the native Aβ monomer. In contrast, without the photothermal effect, some native Aβ monomers will form aggregates, which greatly reduce the photo-oxygenation efficiency due to the reduced surface area exposed to singlet oxygen and the additional energy required to dissociate the Aβ aggregates. Therefore, the combination of photothermal and photo-oxygenation treatments can generate synergistic effects to largely enhance the inhibition efficiency of Aβ_42_ aggregation compared with a single modality of photo-treatment. The degree of Aβ_42_ oxygenation could be used to evaluate long-term photo-inhibition capability. For this purpose, mass spectroscopy (MS) analysis was used to analyze the formation of oxygenated adducts in the protein. Native Aβ_42_ monomers showed an intense signal of 4514 Da in the MS spectrum, which corresponded to the molecular weight of the native Aβ_42_ monomer (Figure 5c) [61]. When the Aβ_42_ monomers were treated with NIR-activated PCN−222@ICG, without and with photothermal effects, under NIR irradiation (0.025 mg mL^−1^, 0.6 W cm^−2^) for 5 min, and then, incubated for 0.5 h, a new peak representing an oxygenated adduct with a higher molecular weight at 4538 Da was observed in the MS spectra (Figure 5d,e). It was observed that the PCN−222@ICG group excluding the photothermal effect had less oxygenated Aβ compared with the PCN−222@ICG group with the photothermal effect; this further demonstrated our hypothesis that the photothermal effect could enhance the photo-oxygenation of Aβ. The combination of photothermal treatment and photo-oxygenation would generate a synergistic effect to effectively inhibit Aβ_42_ aggregation with a long-term effect.

### 2.6. Attenuation of Aβ-Induced Cytotoxicity by Photo-Activated PCN-222@ICG

We then investigated the attenuation of Aβ-induced cytotoxicity by photo-activated PCN−222@ICG using an in vitro PC12 cell model. The PC12 cell line is a classical neuronal cell model for studying Aβ-induced cytotoxicity [62]. The intrinsic biocompatibility of PCN−222 and PCN−222@ICG were first investigated via CCK-8 assay on PC12 cells. After 24 h of incubation with PCN−222 and PCN−222@ICG, the cell viability of the PC12 cells remained over 90% at a concentration as high as 200 µg mL^−1^, indicating the low cytotoxicity of PCN−222 and PCN−222@ICG (Figure 6a). When co-cultured with PC12 cells, PCN−222@ICG at various concentrations showed low cytotoxicity for up to 72 h (Appendix A). We then investigated the attenuation of Aβ-induced cytotoxicity for the five groups of PC12 cells with Aβ_42_ in the absence of nanoprobes, in the presence of PCN−222 (25 µg mL^−1^) without NIR irradiation, in the presence of PCN−222 (25 µg mL^−1^) with NIR irradiation (0.6 W cm^−2^ for 30 min), in the presence of PCN−222@ICG without irradiation, and in the presence of PCN−222@ICG (25 µg mL^−1^) with NIR irradiation (0.6 W cm^−2^ for 30 min), respectively (Figure 6b). Four groups of PC12 cells without Aβ_42_ were used as control groups, including only PC12 cells, PC12 cells with NIR irradiation, PC12 cells with PCN−222 nanoprobes and PC12 cells with PCN−222@ICG nanoprobes. As shown in Figure 6b, the cell viability of all four control groups exceeded 90%. The group of PC12 cells with Aβ_42_ in the absence of nanoprobes showed a low cell viability of ~30% due to the obvious cytotoxicity of the Aβ aggregates. The addition of PCN−222 and PCN−222@ICG to PC12 cells in the presence of Aβ_42_ without NIR irradiation showed a slight increase in cell viability to ~40%. This could be explained by the adsorption of some Aβ_42_ monomers on the nanoprobe surface which prevented the absorbed Aβ_42_ monomers from forming neurotoxic Aβ_42_ aggregates. Photo-activated PCN−222 and photo-activated PCN−222@ICG nanoprobes could recover cell viability to around ~70% and ~80%, respectively, which indicated that oxygenated Aβ_42_ would no longer aggregate into toxic high-order structures and remained non-toxic. PCN−222@ICG nanoprobes showed especially higher cell viability compared to PCN−222@ICG, which demonstrated that photothermally assisted photo-oxygenation had a higher capability for the attenuation of Aβ-induced cytotoxicity. Florescence live/dead staining was also carried out to analyze the attenuation of Aβ-induced cytotoxicity for PC12 cells in the presence of Aβ_42_ with treatment using PCN-222 and PCN−222@ICG under NIR light irradiation (Figure 6c). Obviously, the cells with Aβ_42_ after treatment with photo-activated PCN−222@ICG showed a higher ratio of live cells with green color compared to the PCN−222 group. In contrast, the control group of PC12 cells without nanoprobe treatment showed numerous dead cells with red color. The above results demonstrate that photo-activated PCN−222@ICG had a higher ability to attenuate Aβ-induced cytotoxicity.

### 2.7. In Vitro Brain-on-a-Chip Model for BBB Permeability Test of RVG-Modified PCN-222@ICG

Drugs that are designed for brain diseases should be capable of crossing the BBB to enter the central nervous system (CNS). In order to achieve better BBB permeability for PCN−222@ICG, we modified the nanoprobe with an RVG peptide, which is widely used for the transvascular delivery of small interfering RNA to the CNS [63]. To determine whether the PCN−222@ICG@RVG nanoprobe can penetrate the human BBB, we fabricated a microfluidic brain-on-a-chip device using primary human cells with continuous flow that simulated the microenvironment of the human brain.

This brain-on-a-chip was composed of two polymeric polydimethylsiloxane (PDMS) compartments with channels (the upper layer for the brain part: 1mm wide and 1mm high; the lower layer for blood vessel: 1 mm wide and 0.2 mm high) separated by a thin and porous PET membrane (0.4 µm pore, 4 × 10^6^ pores cm^−2^), fixated with a silicone sealant (Figure 7a). The membrane was sandwiched between two PDMS compartments after treatment with oxygen plasma at 35 W for 1 min. Each PDMS compartment was connected to an inlet channel and an outlet channel for solution filling, respectively. Figure 7b shows the fabricated brain chip filled with blue and red ink representing the upper and lower compartments and channels, respectively. To establish the BBB model, brain endothelial phenotype HUVECs derived from human temporal lobe microvessels were first seeded on the lower surface of the membrane and on the surface of the lower compartment to mimic the wall of brain microvessel. When the endothelial cellular monolayer was formed, the device was turned over to seed human astrocytes on the other side of the membrane in the upper compartment (brain compartment) to form a cell co-culture brain chip model (Figure 7c). The chip could be connected to a peristaltic pump with a continuous flow rate of 60 μL h^−1^ to provide dynamic culture conditions mimicking the brain’s physiological environment (Appendix A). Here, reservoirs containing endothelial medium and nanoprobes were connected to the inlets of the lower channels (blood side), while astrocyte medium flowed into the upper channels (brain side). The effluent was then collected using Eppendorf tubes on the other side of the peristaltic pump. To let the cells firmly attach to the membrane of the chip, an extracellular matrix consisting of collagen IV (400 μg mL^−1^) and fibronectin (100 μg mL^−1^) was coated on both channels of the chip. Twenty-three million HUVECs were detached from cell culture plates and seeded on the bottom channel of the microfluidic chip in the endothelial medium. For the co-culturing of astrocytes in the brain channel of the chip, a density of 0.7 × 10^6^ cells mL^−1^ of human astrocyte in the astrocyte media was seeded on the apical channel of the chip. After 24 h of incubation, confluent monolayers of astrocytes and endothelial cells were formed on the brain side and blood side under the continuous flow environment, respectively (Figure 7d,e). An immunocytochemistry image of the co-cultured HUVECs with the astrocytes in the whole brain chip is shown in Figure 7f. The results show that HUVECs stained with the phalloidin formed a uniform monolayer on the blood side, while the astrocytes expressing GFAP formed an intricate network within the brain compartment (Figure 7g,i). The high-level expression of the tight junction protein ZO-1 of HUVECs was also observed, which demonstrated the formation of the tight junction of the BBB to prevent the leakage of transported solutes and water (Figure 7h). The cell–cell tight junctions of HUVECs were critical for mimicking the in vivo-like BBB.

To examine the organ-level functionality of our brain-on-a-chip model, neuroinflammation was mimicked by perfusing tumor necrosis factor α (TNF-α) (100 ng mL^−1^) through the blood compartment for 12 h. The immunocytochemistry images showed that the response to inflammatory stimulation induced by TNF-α reduced the expression of the tight-junction marker ZO-1 (Figure 8a,b). To investigate the barrier integrity of our BBB model, the fluorescently labeled dextran tracers (3 k, 10 k, and 70 kDa) were perfused into the blood channel for 12 h and the permeability of these agents through the BBB layer was accessed via fluorescence spectroscopy. The results show that the apparent permeability (*P_app_*) of the BBB model without TNF-α treatment was generally low with an inverse correlation with the size of the tracers (average *P_app_* = 1.55 × 10^−7^ cm s^−1^, 1.98 × 10^−8^ cm s^−1^, and 3.45 × 10^−9^ cm s^−1^ for 3 k, 10 k, and 70 kDa dextran, respectively) (Figure 8c). In contrast, more dextran leaked to the brain side with the treatment of TNF-α, leading to a large increase in *P_app_* of 1.8 × 10^−6^ cm s^−1^, 2.25 × 10^−7^ cm s^−1^, and 3.41 × 10^−7^ cm s^−1^ for 3 k, 10 k, and 70 kDa dextran, respectively. After the damage to the endothelium caused by the TNF-α-induced inflammation, the permeability of 70 kDa dextran increased to a similar level to that of 10 kDa dextran (Figure 8d). The relative ZO-1 expression displayed a concentration-dependent decrease after treatment with TNF-α (Figure 8e), which was consistent with the immunofluorescence microscopic analysis in Figure 8a,b. These results demonstrate that this brain chip simulated an in vivo-like brain microenvironment and functions that resemble the interface of the neurovascular unit.

To investigate the BBB-penetrating capacity of nanoprobes via RVG conjugation, PCN−222@ICG nanoprobes with and without RVG of the same concentration were dosed into the blood side in the endothelial cell culture media. The blood–brain barrier integrity was then measured using fluorescently labeled dextran tracers (70 kDa). As shown in Figure 8f, nanoprobes without RVG showed a *P_app_* of 5.29 × 10^−8^ cm s^−1^ after 12 h of flowing. In contrast, RVG-modified PCN−222@ICG nanoprobes exhibited approximately a 2.6 times higher permeability of 1.37 × 10^−7^ cm s^−1^ after 12 h of flowing compared with the unmodified nanoprobe, which indicated that RVG modification could efficiently assist in the translocation of nanoprobes through the human BBB. It has been reported in other studies that fluid flow is also important to sustain barrier function over time compared with a static environment [46]. Here, an in vitro static BBB Transwell model was created and used as a control for the BBB chip study. Human HUVECs and human astrocytes were cultured on the upper side and the lower side of the 0.4 μm-pore-size membrane in the BBB Transwell, respectively (Appendix A). FITC-labeled dextran tracers (3 kDa, 10 kDa, and 70 kDa) were used to determine the barrier integrity. It was demonstrated that the translocation rate of 3 kDa dextran was 1.23 × 10^−6^ cm s^−1^, around 2 times higher than that of 10 kDa, which was 6.48 × 10^−7^ cm s^−1^, and nearly 3 times higher than that of 70 kDa dextran, the permeability of which was 4.15 × 10^−7^ cm s^−1^ (Appendix A). The results indicate that the static BBB Transwell exhibited similar barrier integrity and function compared with those of the microfluidic brain chip. The PCN−222@ICG nanoprobes with and without RVG conjugation were tested using this static BBB Transwell model. The translocation rate of PCN−222@ICG@RVG was 4.8 × 10^−7^, which was 2.42 times higher than that of unconjugated nanoprobes after 12 h (Appendix A). The results were consistent with those from the chip model studies in that RVG conjugation could largely increase the BBB permeability of PCN-222@ICG nanoprobes.

### 2.8. Ex Vivo Dissociation of Aβ Plaques on AD Mouse Brain

In the AD brain, particularly in the late-onset AD brain, Aβ fibrils are prone to aggregate and form Aβ plaques, which are neurotoxic to brain cells, including neurons and astrocytes [64,65,66,67]. To investigate whether the NIR-activated PCN−222@ICG@RVG nanoprobe is capable of inhibiting and disassembling high-order Aβ plaques, a transgenic AD mouse brain containing Aβ plaques was sliced and fixed on a glass slide, followed by treatment with NIR-activated PCN−222@ICG@RVG (Figure 9a). The Aβ-containing brain slice was stained using ThS, a fluorescent dye that exhibits high fluorescent emission when binding with amyloid plaques. The fluorescent images of the brain slice with different treatments, including NIR-only treatment (0.6 W cm^−2^), NIR treatment with PCN−222@RVG nanosheets (0.2 mg mL^−1^), and NIR treatment with PCN−222@ICG@RVG nanoprobes (0.2 mg mL^−1^), were taken to compare the Aβ plaque degree. It was obvious that NIR light alone did not affect the Aβ plaques in the brain slice, whereas PCN−222@RVG with NIR irradiation treatment and PCN−222@ICG@RVG with NIR irradiation resulted in less Aβ plaque in the brain slice (Figure 9b). Moreover, the PCN−222@ICG@RVG-with-NIR irradiation group showed a significant decrease in the fluorescence density of Aβ plaques, even compared with the PCN−222@RVG-under-NIR irradiation group; this indicates that NIR-activated PCN−222@ICG@RVG had an enhanced dissociation effect on the Aβ plaques due to the combined photo-treatment. Furthermore, to quantitatively analyze Aβ plaques in different treatments, Image J software was used to analyze the density of Aβ plaques in the ex vivo brain slice. The only NIR-treated group brain slice contained Aβ plaques with a density of 59.2 per cm^2^, which was similar to that of the non-treated group (61.7 per cm^2^). In contrast, the NIR-stimulated PCN−222@RVG group had a lower density of Aβ plaques at 25 per cm^2^. NIR-activated PCN−222@ICG@RVG had a remarkably low density of Aβ plaques, at only 9.3 per cm^2^. Moreover, the brain slice treated with NIR-activated PCN−222@ICG@RVG also exhibited much smaller Aβ plaques than the two control groups and the NIR-excited PCN−222@RVG group. This result further evidenced the enhanced Aβ plaques’ dissociation efficacy due to the combined phototherapy effects.

## 3. Discussion

In recent years, many groups have aimed to use nanoparticle-based phototherapy approaches to advance the treatment of Alzheimer’s disease [28,32]. The main difficulty of using phototherapy for brain treatment is the low penetration of light into the brain since most current nanoparticle-initiated phototherapy approaches are based on visible light. Photo-oxygenation for the inhibition of Aβ_42_ aggregation has been reported, but it is limited by the relatively low inhibition effects. In this study, we have developed a PCN−222@ICG nanoplatform which showed a quick response of temperature change under 808 nm NIR irradiation. The temperature of nanoparticles can rapidly increase from 25 °C to 45 °C after 5 min of laser irradiation, even at a low concentration. Through rigorous characterization of PCN−222@ICG@RVG, we found that these nanoparticles could generate not only a photothermal effect but also singlet oxygen upon NIR irradiation, both of which contributed to inhibiting Aβ_42_ aggregation. The Aβ_42_ aggregation degree was characterized by various means, including DLS, electro-microscopy, circular dichroism, and protein gel electrophoresis. The photothermal effect of PCN−222@ICG can efficiently enhance the photo-oxygenation based inhibition of Aβ_42_ aggregation. Therefore, the combination of photothermal and photo-oxygenation treatments displayed a synergistic effect for enhancing the inhibition efficiency of Aβ_42_ aggregation compared with a single modality of photo-treatment. The cytotoxicity of Aβ_42_ aggregates can be significantly reduced in the presence of PCN−222@ICG and NIR irradiation, suggesting that photo-activated PCN−222@ICG can rescue the neuron cells which are exposed to neurotoxic Aβ aggregates.

We further leveraged organ-on-chip technology to test whether PCN−222@ICG with the conjugation of RVG can penetrate the blood–brain barrier and CNS. PCN−222@ICG appeared to be able to cross the blood–brain barrier lined with HUVEC, human primary astrocytes on a dual-channel microfluidic chip. This study also shed light on future brain drug screening using in vitro blood–brain barrier models. The comparison of mouse brain slices showed a striking difference between the treated and non-treated groups in terms of the number of Aβ aggregates under ThS staining. Taken together, we believe that this PCN−222@ICG with the conjugation of RVG can reduce the neurotoxicity of Aβ by inhibiting the amyloid from aggregating into toxic forms.

There are certain limitations for this study. The quantitative measurement for Aβ aggregation during nanoparticle treatment was lacking. Although the ThT assay could indicate the aggregation of the peptide, the accurate structures of oligomers and fibrils were not determined using this method. Furthermore, despite the fact that the ex vivo AD mouse brain slice was used to study the inhibition effect of PCN−222@ICG@RVG on Aβ aggregation, no in vivo animal experiments were performed to further demonstrate the inhibition effects in AD mouse models. This will be included in our future work.

## 4. Materials and Methods

### 4.1. Chemicals

The Aβ_42_ peptide was purchased from GL Biochem Ltd. (Shanghai, China). Thioflavin T; 5, 10, 15, and 20-tetrakis (4-carboxyphenyl)porphyrin (TCPP); zirconyl chloride octahydrate (ZrOCl_2_·8H_2_O); formic acid (FA); and indocyanine green (ICG) were purchased from Tokyo Chemical Industry Co., Ltd. (Tokyo, Japan). Dimethyl sulfoxide (DMSO, anhydrous, >99.9%) and N, N’-dimethylformamide (DMF, anhydrous, 99.9%) were purchased from J&K Scientific Ltd. (Beijing, China). All reagents and chemicals were purchased commercially and without further purification if not otherwise mentioned, and Ultrapure Milli-Q water (18.2 MΩ) was used in all experiments. HEPES buffer (pH = 7.4) was purchased from Thermo Fisher Scientific Inc. (Waltham, USA). The native gel electrophoresis-related chemicals and kits were purchased from Bio-Rad Laboratories, Inc. (Hercules, USA). All other reagents were purchased from Sigma-Aldrich (St. Louis, USA) unless otherwise specified.

### 4.2. Synthesis of 2D PCN-222 Nanosheets

Two dimensional PCN−222 nanosheets were prepared according to the previous method with slight modifications [68]. Briefly, 10 mg ZrOCl_2_·8H_2_O, 30 mg TCPP, 240 µL of FA, and 50 µL water were added to DMF to a final volume of 2 mL in a Teflon-lined stainless-steel autoclave; then, they were heated at 120 °C for 24 h. After cooling down to room temperature, the PCN−222 nanosheets was separated through centrifugation at 13,500 rpm for 30 min and further rinsed with DMF and ethanol several times until there was no red fluorescence in the supernatant under UV light. A total of 50 mg as-synthesized MOFs and 1 mL 8 M HCl were added into 20 mL DMF, and the DMF suspension was stirred at 120 °C for 12 h to remove unreacted inorganic species, starting ligands, and modulating reagents. The products were further separated via centrifugation, and then, dispersed into fresh acetone for 24 h to exchange and remove the DMF. After the removal of acetone via centrifugation, the sample was activated by drying it under vacuum at 100 °C for 24 h.

### 4.3. Preparation of PCN-222@ICG

Fifty-milligram PCN-222 nanosheets were first sonicated for 5 min and mixed with 10 µM of ICG in ultrapure water and stirred for 24 h at room temperature. Residual ICG was then removed via centrifugation at 13,500 rpm for 30 min and rinsed with water several times.

### 4.4. Preparation of RVG-Conjugated PCN-222@ICG

For the immobilization of the RVG peptide in PCN−222@ICG via electrostatic adsorption, the obtained PCN−222@ICG was mixed with 500 µL 5 mg mL^−1^ RVG peptide and shaken at 4 °C for 4 h. Thereafter, the product was centrifuged at 12,000 rpm for 10 min and washed with Milli-Q water three times.

### 4.5. Characterization of PCN-222@ICG Nanoprobes

Transmission Electron Microscopy (TEM) images were collected using a JEOL JEM-2100F TEM microscope. Scanning Electron Microscopy (SEM) images were obtained using a JEOL Field Emission SEM microscope. The ultraviolet–visible (UV-vis) absorbance levels of PCN−222 and PCN−222@ICG were measured using an Ultrospec 2100 Pro spectrophotometer (Amersham Biosciences). Powder X-ray diffraction (PXRD) patterns were collected at 293 K using a Rigaku SmartLab X-Ray diffractometer via Cu Kα radiation. The zeta potential and size distribution of the PCN−222@ICG nanosheets were measured using a Malvern ZEN 3600 Zetasizer. The Fourier Transform Infrared (FT-IR) spectra were obtained using a Bruker Vertex-70 IR Spectroscopy in the wavelength region of 400–4000 cm^−1^. Brunauer–Emmett–Teller (BET) surface area analysis was performed using a Micromeritics ASAP 2420 analyzer.

### 4.6. Characterization of Photothermal Effect

PCN−222@ICG at various concentrations was dispersed in water in a 1.5 mL centrifuge tube, which was then exposed under an 808 nm light at a power density of 0.6 W cm^−2^ in a black container. Temperature change was measured using a FLIR C2 thermal imaging camera at different time points.

### 4.7. Singlet Oxygen Generation Measurement

Singlet oxygen was characterized using a DPBF assay. Briefly, PCN−222@ICG (0.2 mg mL^−1^) and DPBF (0.04 μM) were dissolved in DMF and stirred for 5 min before exposed to an 808 nm NIR light at a power density of 0.6 W cm^−2^. The UV-vis absorption was recorded using a spectrophotometer at the same time intervals as mentioned in Figure 3a. ^1^O_2_ was also identified via ESR obtained using a JES-FA200 Electron Spin Resonance Spectrometer using 4-oxo-TEMP.

### 4.8. Preparation of Monomeric Aβ_42_ Solution

Human Aβ_42_ (1 mg) was dissolved in hexafluoro-2-propanol (HFIP) and kept at room temperature for 2 h. The HFIP was then removed under a gentle flow of nitrogen gas, followed by freeze-drying for 3 h. The obtained Aβ_42_ peptide (1 mg) was dissolved in 10 µL DMSO, and then, diluted to 300 µM with ultrapure water (Invitrogen). The solution was further diluted in HEPES buffer (20 µM, pH 7.4, 150 µM NaCl) to a final concentration of 25 µM for the inhibition study.

### 4.9. Inhibition Study of Aβ_42_ Aggregation under NIR Light Irradiation

For the inhibition studies, PCN−222 nanosheets and PCN−222@ICG were mixed with Aβ_42_ solution (25 µM) at a concentration of 100 µg mL^−1^. The mixed solution was then exposed to NIR light irradiation (808 nm, 0.6 W cm^−2^) for 30 min and incubated at 37 °C for 24 h.

### 4.10. Thioflavin T (ThT) Assay

ThT stock solution was prepared by dissolving 0.32 mg of ThT in 10 mL ultrapure water and filtering through a 0.22 μm PES syringe filter. Ten micromolar of the prepared ThT solution was mixed with 15 µM of Aβ_42_ peptide with/without nanoprobe treatments. The fluorescence was measured using a photoluminescence spectrometer. The fluorescence intensity was recorded using an excitation wavelength of 440 nm and an emission wavelength of 485 nm. All sample measurements were performed in triplicate. PCN−222 nanoprobes were removed via centrifugation (21120 g, 30 min) before being subjected to ThT assay to prevent the nanoparticles from interfering with the photoluminescence property of ThT during the analysis of Aβ_42_ aggregation.

### 4.11. Circular Dichroism (CD) Measurement

The far-UV (190–250 nm) of the CD spectra were recorded with a JASCO J-810 Spectrometer (JASCO Co., Tokyo, Japan), using a quartz cuvette with 1 mm path length. A total of 25 µM of Aβ_42_ peptide with and without nanoprobe treatments was used for CD measurement. The CD spectrum was scanned three times in the range of 195–250 nm under a N_2_ blowing atmosphere. Conformational changes in the peptides were analyzed at 195 and 216 nm, respectively.

### 4.12. Dynamic Light Scattering (DLS) Measurement

The size distribution of the incubated Aβ_42_ peptide (25 µM) was measured using a light scattering spectrometer (Zetasizer Nano ZS instrument; Malvern Instruments, Worcestershire, UK). The Aβ_42_ samples were centrifuged at 16,000× *g* for 30 min at 4 °C and the supernatant was measured via DLS at 25 °C. Each scan consisted of 11 runs and each sample was measured at least three times.

### 4.13. Transmission Electron Microscope (TEM) Analysis

Five microliters of each Aβ_42_ sample (25 µM) were placed on a glow-discharged, 300-mesh formvar/carbon coated copper grid. The samples were stained with 2% uranyl acetate, and then, the grids were washed at least three times, followed by drying at room temperature before measurement. The TEM images were collected using a JEOL JEM-2100 F microscope with an accelerating voltage of 75 kV.

### 4.14. Native Gel Electrophoresis Analysis

The Aβ_42_ samples with various treatments were mixed with the sample buffer and loaded on a 10% precast gel (SDS free, Bio-Rad Laboratories, Inc., Hercules, USA) for measurement in a MiniPROTEAN^®^ Tetra Cell (Bio-Rad Laboratories, Inc., Hercules, USA) with a current of 100 V for 80 min. Silver staining was performed using a silver staining kit (Bio-Rad Laboratories, Inc., Hercules, USA) according to the manufacturer’s instructions. The gel was then imaged using a ChemiDoc Touch Imaging System (Bio-Rad Laboratories, Inc., Hercules, USA).

### 4.15. Cell Counting Kit 8 (CCK-8) Assay

PC 12 cells were cultured in medium containing 15% HS, 2.5% FBS, and 1% antibiotics under a 5% CO_2_ atmosphere at 37 °C. For the biocompatibility test, around 0.5 × 10^4^ cells per well were seeded into a 96-well plate and incubated for 24 h. PCN−222 nanosheets were then added to the 96-well plate at different concentrations and incubated for 24 h before being subjected to the CCK8 assay according to the manufacturer’s protocol. For the photo-inhibition test, non-treated, NIR light-treated, PCN−222-treated, and PCN−222-with-light-illumination-treated Aβ42 peptides were added to the PC 12 cells in a 96-well plate and incubated for 24 h before being subjected to the test.

### 4.16. MALDI-TOF MS Measurement

MALDI-TOF MS spectra were recorded via Bruker UltrafleXtreme MALDI-TOF-TOF Mass Spectrometer using α-cyano-4-hydroxy cinnamic acid as a matrix. Native Aβ_42_, and Aβ_42_ irradiated with NIR light in the presence of PCN−222@ICG for 0.5 h and 1 h were measured using the MALDI-TOF MS.

### 4.17. BBB-on-a-Chip Set-Up

BBB chips were fabricated using a 3D-printed mold. A porous PET membrane (0.4 μm pores, density of 4 × 10^6^ cm^−2^) was sandwiched between the two PDMS microfluidic channels during bonding. The human primary umbilical vein endothelial cell (HUVEC) was propagated on tissue-culture plates that were coated with Matrigel using extracellular matrix (ECM) media and maintained according to protocols provided by Thermo Fisher Scientific Inc. (Waltham, USA). Primary human astrocytes isolated from the cerebral cortex were obtained from ScienCell and maintained in the Astrocyte medium. The primary cells were used at passages 3–6. The microfluidic channels were coated with ECM consisting of collagen IV (400 μg mL^−1^) and fibronectin (100 μg mL^−1^) overnight. Both channels of the chip were rinsed with PBS three times, and then, filled with astrocyte media for at least 1 h before seeding cells. For the co-culturing of astrocytes in the brain channel of the chip, a density of 0.7 × 10^6^ cells mL^−1^ of human astrocyte were mixed together in the astrocyte media and seeded on the apical channel of the chip; then, they were incubated in the incubator for 1 h. To remove the access of the astrocyte, the channels of the chip were washed with the endothelial medium. Afterwards, 2.3 × 10^7^ cells mL^−1^ of HUVECs were seeded in the basal channel, and the device was flipped immediately to allow the HUVECs to adhere to the ECM-coated PET membrane. After 5 h of incubation, the device was flipped back to let the rest of HUVECs sit on the bottom and sides of the channel to form a capillary lumen. After 12 h of incubation, the chip was connected to a peristaltic pump and endothelial medium was allowed to flow through the channels at a flow rate of 60 μL h^−1^ to allow the chip to adjust to the flow conditions. FITC-labeled dextran tracers were dosed through the vascular channels for a known period of time, and the concentrations of the dextran tracers in the outlet samples from both vascular and brain channels were determined using a Synergy H1 microplate reader.

### 4.18. Apparent Permeability (P_app_) Calculation

The apparent permeability (P_app_) of the blood–brain barrier was calculated according to the following equation, described in previous report [24]:
$$P_{\mathrm{app}}=\frac{Vr\times Cr}{A\times t\times \frac{\left(Cd\times Vd+\mathrm{Cr}\times \mathrm{Vr}\right)}{Vd+Vr}}$$
where *V_r_* is the volume of the receiving channel effluent after time *t*; *V_d_* is the volume of the dosing channel effluent after time *t*; *A* is the area of membrane; *C_r_* is the measured concentration of the tracer in the receiving channel effluent; and *C_d_* is the measured concentration of the tracer in the dosing channel effluent.

### 4.19. Transwell Model of BBB

The BBB Transwell model was created via a contact dual-culture method. Specifically, 1 × 10^5^ astrocytes were seeded on the lower side of the Transwell membrane by flipping the inserts of the Transwell. After the attachment of the astrocyte to the membrane for around 1 h, 1 × 10^6^ HUVECs were then seeded on the upper side of the Transwell membrane and kept at 37 °C for 12 h before use. Fluorescence-labeled dextran tracers (3 k, 10 k, 70 kDa) were dosed into the upper chamber of the Transwell. The fluorescence intensity of the dextran tracers and the nanoprobes in both the blood and brain chambers were measured using a microplate reader. The apparent permeability of the different molecular size tracer and nanoprobes (*P*) was calculated as:$$P=\frac{1}{C_{0}A}\frac{dQ}{dt}$$
where *C*_0_ is the dosing concentration, *A* is the total surface area of the Transwell membrane, and *dQ*/*dt* is the transport rate calculated as the gradient of mass over time.

### 4.20. Ex Vivo Evaluation

For ex vivo evaluation of the therapeutic effect of PCN@ICG on the dissociation of Aβ_42_ plaques, brain tissue containing Aβ plaques was extracted from the AD mouse model (APP, aged 9 months). The extracted brain was sliced in 30 μm-thick slices in their frozen state using a cryostat (JJQ-L2016, Wuhan Junjie Electronic Ltd., Wuhan, China) and fixed on a glass slide. A drop of PCN@ICG nanosheets (0.025 mg mL^−1^) was added to the brain tissue on a glass slide. After 4 h of incubation in a refrigerator, the glass slide was placed under 808 nm NIR irradiation (0.6 W cm^−2^) for 10 min. The brain slice was then stained with 1 mM Thioflavin S (ThS) solution for 30 min, followed by washing with PBS solution three times. The brain slice was imaged under a confocal microscope (Leica TCS SPE, Leica Microsystems Ltd., Wetzlar, Germany) to take fluorescence images. The above animal experiment has obtained animal ethics approval from the Animal Subjects Ethics Sub-Committee (ASESC) of the Hong Kong Polytechnic University (Approval No: 20-21/6-BME-R-CRF).

## 5. Conclusions

In summary, we have demonstrated enhanced inhibition and dissociation of Aβ aggregates and good human BBB permeability using an NIR-activated PCN−222@ICG hybrid nanoprobe. With the conjugation of an RVG peptide, we observed that the conjugated nanoparticles could penetrate the BBB via an in vitro brain-on-a-chip model. The Aβ inhibition effect of this nanoparticle was also confirmed in an ex vivo study. Local photothermal heating could increase the instability of Aβ_42_ aggregation to facilitate the photo-oxygenation of the Aβ_42_ monomer, leading to a synergistic inhibition effect which could not be achieved by a single modality of photo-treatment. The successful demonstration of largely enhanced Aβ plaque inhibition and dissociation efficacy under moderate NIR laser irradiation via a combinational phototherapy approach, using in vitro brain-on-a-chip and ex vivo evaluation, is critical for future in vivo phototherapy of AD; this is because NIR light may decay during transmission to the brain through the skull. Moreover, through conjugation with the brain-penetrating peptide RVG, this hybrid nanoprobe shows good human BBB permeability in a human brain-on-a-chip, which allows for nanoprobe-based in vivo brain therapy. This study may shed light on the future development of nano-therapeutics for Alzheimer’s disease and other brain disorders that need to be BBB-penetrative, and provides a new paradigm for potent clinical treatment for AD patients.

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
