# Peer review of "Near-Infrared Photothermally Enhanced Photo-Oxygenation for Inhibition of Amyloid-β Aggregation Based on RVG-Conjugated Porphyrinic Metal–Organic Framework and Indocyanine Green Nanoplatform"

_ijms, 2022, doi:10.3390/ijms231810885_

Round 1
Reviewer 1 Report
Alzheimer’s disease is a progressive neurodegenerative disease with no effective cure so far. Recently, photo-oxygenation was used to induce the clearance and suppression of aggregation of amyloidogenic proteins. In this manuscript, the authors investigate PCN-222@ICG photo-oxygenation nanosheets to inhibit Aβ42 aggregates as a potential therapeutic strategy for AD. This is a very interesting study, however, it needs a carful revision. Below are my comments explained point by point.
Major issues:
1. The main issue is that the authors investigated the inhibition effect of PCN-222@ICG nanosheets and not (PCN-222@ICG@RVG) nanoprobes, so we do not know if the RVG-conjectured PCN-222@ICG nanoprobes could or have the same inhibition efficiency as PCN-222@ICG nanosheets?
2. The title of the manuscript is so long, it needs to be shortened, also should be changed to precisely reflect the work.
3. Scheme 1 should be modified to reflect what has been done here in this study. See comment number 1.
4. The authors state that “the photo-oxygenation process generates oxidized Aβ monomer with low aggregation capability” have they detected the oxidation modification of Aβ peptide?
5. Add references to support your claims/statements in page2, lines 63-65
6. In the introduction, the authors introduced RVG briefly with no references, please elaborate supported by references.
7. In Figure 1c, the UV-vis spectrum for ICG alone should be added.
8. Why the authors did not used ThT assay to monitor the Aβ42 fibrillation kinetics and to evaluate the inhibition effect of (PCN-222@ICG? ThT data must be provided. TEM images of Aβ samples at zero hour should be provided as well.
9. In Fig 4e, the CD spectra at zero h should be provided. Also, why both PCN-222 and PCN@ICG-treated Aβ samples did not show random coil signal after 24 h incubation? it seems that treated samples have conformation different from that at zero h, which what it expected to be random coil?
10. In fig 5a, in the ThT data, have the authors checked the possible fluorescence quenching by PCN-222 and PCN-222@ICG?
11. Fig 5c is not clear, add some labelling/description and make it bigger. Also, the font style is different.
12. In page 10, line 312-313, the authors state that “Once the photothermal treatment stops by turning 312 off of laser irradiation, native Aβ monomers will aggregate again”, what is the evidence for this?
13. In cytotoxicity assay what was the concentration for Aβ42?
14. In Fig 9b, again have the authors checked the possible quenching effect for ThS intensity by the nanoprobes? High resolution images (TEM images) for brain sections showing the Aβ plaques in all groups should be provided.
15. The conclusion should be rewritten, the authors stated that “we have demonstrated enhanced inhibition and dissociation of Aβ aggregates and good human BBB permeability using NIR-activated PCN-222@ICG@RVG hybrid nanoprobe” this claim is not supported by evidence, as they only tested the PCN-222@ICG nanosheets for inhibition/disassembly effect.
Minor issues
1. Delete “Alzheimer’s” in Alzheimer’s Aβ aggregation
2. No need to repeat the abbreviations
3. Page 2, line 46, reference number 5 has different citation style
4. Page 8, line 263, PCN-222@ICG (0.025 mg mL-1), is duplicated, delete one of them
5. Page 10, line 332, “in vitro” should be italic. Also, please check the font size in the same paragraph.
6. Page 14, line 503, “Aβ plagues” should be Aβ plaques. Same in page 19, line 696
7. Page 15, line 526, the reference of the original method should be added.
8. Page 16, line 566, 0.2 mg mL-1 of what?
Author Response
Please refer to the attached response letter

Reviewer 2 Report
16 August 2022
Regarding the review of manuscript ‘Near-Infrared Photothermally-Enhanced Photooxygenation for Inhibition of Alzheimer’s Amyloid-β Aggregation and Attenuation of Neurotoxicity Based on Brain-Targeting Peptide RVG conjugated Two-dimensional Porphyrinic Metal-Organic Framework@Indocyanine Gre’ by Wang J et al., submitted to International Journal of Molecular Sciences (IJMS)
Manuscript ID: ijms-1881132
Dear Authors,
In this original article entitled ‘Near-Infrared Photothermally-Enhanced Photooxygenation for Inhibition of Alzheimer’s Amyloid-β Aggregation and Attenuation of Neurotoxicity Based on Brain-Targeting Peptide RVG conjugated Two-dimensional Porphyrinic Metal-Organic Framework@Indocyanine Gre’, Wang and colleagues investigated the effects of a new photothermally assisted photooxygenation treatment (PCN-222@ICG@RVG) on amyloid-beta (Aβ) aggregation.
The main strength of this original literature review is that it addresses an interesting and innovative technique, presenting that photothermally assisted photooxygenation inhibits Aβ aggregation in cell-free and in vitro, that PCN-222@ICG@RVG shows high blood-brain barrier permeability on a chip platform, and that near infrared-activated PCN-222@ICG@RVG dissembles Aβ plaques ex vivo.
In general, I think the idea of this review is really interesting and the authors’ fascinating observations on this timely topic may be of interest to the readers of International Journal of Molecular Sciences. However, some comments, as well as some crucial evidence that should be included to support the author’s argumentation, needed to be addressed to improve the quality of the manuscript, its adequacy, and its readability prior to the publication in the present form, in particular reshaping parts of the introduction and discussion sections by adding more evidence and theoretical constructs.
Please consider the following comments:
1. Abstract: I recommend that the authors add a brief background on the technology leading to the methods.
2. Keyword: I suggest listing ten keywords and use as many keywords as possible in the first two paragraphs of the abstract.
3. In general, I recommend authors to use more evidence to back their claims, especially in the Introduction of the paper, which I believe is currently lacking. Thus, I recommend that the authors attempt to deepen the subject of their manuscript, as the bibliography is too concise: nonetheless, in my opinion, less than 60-70 articles for a research article are really insufficient. I suggest focusing their efforts on researching more relevant literature: I believe that adding more studies and reviews will help them to provide better and more accurate background to this study.
4. Introduction: The section is well-written and nicely presented with a good balance of information on the current development of phototherapy approach to Aβ plaques. However, I recommend that more information on structural and functional changes associated with neurodegeneration in Alzheimer’s diseases (AD) will provide a better and more accurate background. Thus, I suggest the authors to make such effort to provide a brief overview of the pertinent published on neurobiological signs of AD, because as it stands, this information is not highlighted in the text. In this regard, I would recommend citing a recent review that examined pathophysiological basis and biomarkers of AD pathology and investigated molecular signs of neuroinflammation in neurodegenerative diseases, in particular Alzheimer’s disease (https://doi.org/10.3390/ijms21072431; https://doi.org/10.3390/ijms21249338), the involvement of glutamate and GABA; and the influence of the microbiota on the polarization of microglial cells (https://doi.org/10.3390/ijms222111677; https://doi.org/10.3390/ijms221910413; https://doi.org/10.3390/ijms23094476).
5. Introduction: Furthermore, it deserves clearly to state that AD is characterized by cognitive dysfunction, accumulation of Aβ plaques in the cerebral cortex, and hyperphosphorylated tau protein-mediated neuronal damage that is initiated from the hippocampus and cortex region of the brain’, I would suggest adding some studies that discussed amyloid-β (Aβ) pathology in AD and related effect on cognitive abilities, highlighting the combined effect of forms of Aβ and tau protein to drive healthy neurons into the diseased state. Aβ peptide and tau protein consistently accumulate in the frontal and/or parietal lobes, and cause alterations of frontal lobe that impact memory and error-driven learning in individuals who have a high risk of dementia: a novel manuscript provides an overview of the anatomical–functional interplay between the prefrontal cortex and heart-related dynamics in human emotional conditioning (learning) and proposes a theoretical model to conceptualize these psychophysiological processes, the neurovisceral integration model of fear that can be impaired in the context of psychiatric disorder (https://doi.org/10.1016/j.tins.2022.04.003). I believe that adding information from these studies may improve the theoretical background of the present article and its argumentation by highlighting cognitive alterations caused by Aβ plaques in the frontal cortex and prevention of Aβ aggregation as a fundamental approach.
6. Results and Discussion: This section is documented in detail and in depth. Considering the length of this section, I recommend that the authors present separate the section into Results and Discussion. In the discussion section, I suggest that, based on the previous section, the authors fully expand argument presenting the potential, the weakness, and the limitation of this study, the goal, technology needed to achieve this goal, and future research direction, among others.
7. In my opinion, I think the ‘Conclusions’ paragraph would benefit from some thoughtful as well as in-depth considerations by the authors, because as it stands, it is very descriptive but not enough theoretical as a discussion should be. Authors should make an effort, trying to explain the theoretical implication as well as the translational application of their research.
8. References: A review article like this paper needs at least 60-70 citations.
Overall, the manuscript contains no table, nine figures and 33 references. I believe that the manuscript may carry important value sudyingthe effects of a new photothermally assisted photooxygenation treatment on Aβ. I hope that, after these careful revisions, the manuscript can meet the Journal’s high standards for publication. I am available for a new round of revision of this review.
I declare no conflict of interest regarding this manuscript.
Best regards,
Reviewer
Author Response

(The authors gave the same response as above.)

Reviewer 3 Report
This is a nice example of creating a framework for NIR inhibition of Abeta aggregation. The study is well considered, executed, and reported. It will be of interest of the Abeta and AD communities.
1. The authors suggest that the oxidized Abeta monomer has a low aggregation capability. From the ThT data, it looks like the aggregation profile plateaus even in the presence of their construct, but at different intensities. The fibrils also look very different by their TEM images in the presence of PCN-222@ICG, suggesting a different type of fibril is being created in its presence. I suggest the authors address this point in the discussion.
2. The authors should list the concentration of Abeta used in each experiment in the corresponding figure captions. For example, it was strange that there is really no lag phase in Figure 5a, but it makes sense if they used a high concentration of Abeta (25 uM from what I can see in the methods). I could not find what concentration they used in the cell study. Abeta was very toxic in their hands, so I would guess 25 uM.
3. In a future study, it would be worth going to lower concentrations of Abeta (2 uM) and conducting kinetic analyses to see if the mechanism really is based on sequestration or inhibition of the monomer (as done in Heller, Sci Adv, 2020).
4. Molecular markers for reference kDa bands are needed in Figure 4f.
5. Figure 1d is missing numbers on the y-axis, which makes it harder to interpret.
6. The authors need to be explicit in figure caption or the methods how many technical replicates were used in each experiment, and how many experiments were conducted. Figure 6 does not have this information. Figure 9 says n=3, but 3 what? Independent staining procedures? Please be explicit everywhere about what technical replicates represent and how many biological replicates were conducted for the cell and mouse studies.
Round 2
Reviewer 1 Report
The authors have addressed my comments successfully
Reviewer 2 Report
8 September 2022
Regarding the 2nd review of manuscript ‘Near-Infrared Photothermally-Enhanced Photooxygenation for Inhibition of Alzheimer’s Amyloid-β Aggregation and Attenuation of Neurotoxicity Based on Brain-Targeting Peptide RVG conjugated Two-dimensional Porphyrinic Metal-Organic Framework@Indocyanine Gre’ by Wang J et al., submitted to International Journal of Molecular Sciences (IJMS)
Manuscript ID: ijms-1881132
Dear Authors,
I am pleased to see that the authors did an excellent work clarifying most of the comments I have raised in the previous round of the review session. Currently, this paper is a well-written, timely piece of research and investigating the effects of a new photothermally assisted photooxygenation treatment (PCN-222@ICG@RVG) on amyloid-beta (Aβ) aggregation.
Overall, the manuscript contains nine figures, no table, and 68 references. This is a timely and needed work, thus I believe that manuscript now meets the Journal’s standards for publication. I am always available for other reviews of such interesting and important articles. I look forward to seeing further study on this issue by these authors in the future.
I declare no conflict of interest regarding this manuscript.
Best regards,
Reviewer